# The Mitochondrial Genomes of Two Parasitoid Wasps *Protapanteles immunis* and *Parapanteles hyposidrae* (Hymenoptera: Braconidae) with Phylogenetic Implications and Novel Gene Rearrangements

**DOI:** 10.3390/genes14010230

**Published:** 2023-01-16

**Authors:** Dandan Xiao, Ziqi Wang, Jiachen Zhu, Xiaogui Zhou, Pu Tang, Xuexin Chen

**Affiliations:** 1Hainan Institute, Zhejiang University, Sanya 572025, China; 2Guangdong Laboratory for Lingnan Modern Agriculture, Guangzhou 510642, China; 3State Key Laboratory of Rice Biology, Ministry of Agriculture Key Lab of Molecular Biology of Crop Pathogens and Insects, and Zhejiang Provincial Key Laboratory of Biology of Crop Pathogens and Insects, Zhejiang University, Hangzhou 310058, China; 4Institute of Insect Sciences, College of Agriculture and Biotechnology, Zhejiang University, Hangzhou 310058, China; 5Ministry of Agriculture Key Laboratory of Tea Quality and Safety Control, Tea Research Institute of Chinese Academy of Agricultural Sciences, Hangzhou 310008, China

**Keywords:** mitochondrial genome, gene rearrangement, phylogeny, *Protapanteles immunis*, *Parapanteles hypsidrae*

## Abstract

*Parapanteles hypsidrae* (Wilkinson, 1928) and *Protapanteles immunis* (Haliday, 1834) are the most important parasitic wasps of *Ectropis grisescens* Warren and *Ectropis obliqua* (Prout). We sequenced and annotated the mitochondrial genomes of *Pa. hyposidrae* and *Pr. immunis*, which are 17,063 bp and 16,397 bp in length, respectively, and possess 37 mitochondrial genes. We discovered two novel types of gene rearrangement, the local inversion of *nad4L* in *Pa. hyposidrae* and the remote inversion of the block *cox3-nad3-nad5-nad4* in *Pr. immunis,* within the mitogenomes of Braconidae. The phylogenetic analysis supported the subfamily Microgastrinae is a monophyletic group, but the tribes Apantelini and Cotesiini within this subfamily are paraphyletic groups.

## 1. Introduction

Microgastrinae is one of the most numerous subfamilies of the Braconidae with around 3000 known species worldwide. It is present in all of the major terrestrial ecosystems. The majority of the Microgastrinae are lepidopteran koinobiont endoparasitoids. More than 100 species of microgastrines have been utilized in biological control projects, making them one of the most valuable groups in agricultural and forestry pest biological control [1]. Unfortunately, phylogenetic studies on this subfamily are limited and there is still no robust phylogeny for it. The boundaries of several genera are now unclear and occasionally contradictory, and, in addition, future research is likely to modify the current cognitive way of many groups [2,3,4].

*Pa. hypsidrae* and *Pr. immunis* (Hymenoptera: Braconidae: Microgastrinae) are important parasitic wasps of *E. grisescens* and *E. obliqua*, which are two of the most destructive chewing pests in China’s tea plantations. *Pa. hyposidrae* has a black head and mesosoma, a dark brown metasoma with white areas, and a smooth and largely yellowish-green cocoon whereas *Pr. immunis* has a black body with yellow hind femora, and a white cocoon covered with fluffy cotton-like filaments [5]. When the conditions are suitable, the highest natural parasitism rate of tea geometrid larvae by these two parasitoids can reach more than 90%, playing a vital role in the population control of *Ectropis* spp. [5]. Under higher temperatures, *Pr. immunis* outperformed *Pa. hyposidrae*. It was found that parasitism rates decreased with parasitoid density at different temperatures, resulting in less efficient searching [6].

The mitochondrial genome plays a crucial role in phylogenetic construction, and it is inherited from the maternal lineage and cannot be combined with other mitochondrial lineages of insects [7]. The typical insect mitochondrial genome is a circular molecule with a size of 14–19 kb that contains 37 genes, 13 of which are protein-coding genes (PCGs), two of which are ribosomal RNA (rRNA) genes, and 22 of which are transfer RNA (tRNA) genes [8]. The gene arrangement is relatively conservative in insects [9]. However, rearrangement events involving multiple combinations of protein coding genes, tRNA genes and rRNA genes have been observed in Anoplura, Thysanoptera, Corrodentia, Hymenoptera, and other insects [10,11,12,13]. The evolution of gene rearrangements in insect mitogenomes is a hot topic [14]. The rate of gene rearrangement in hymenopteran mitogenomes is extremely high [15,16,17]. Gene rearrangements are typically restricted to specific lineages, which can aid in phylogenetic reconstruction, such as the Braconidae subfamily level [15]. However, there has been very little progress in sequencing the mitochondrial genomes of Microgastrinae. Currently, only sixteen verified mitochondrial genomes of the subfamily Microgastrinae have been reported in GenBank (https://www.ncbi.nlm.nih.gov/; accessed on 12 November 2022), and only two species have all 37 genes identified.

Two mitogenomes of Microgastrinae were newly sequenced using next generation sequencing in this study. We obtained the mitochondrial genomes of *Pa. hyposidrae* and *Pr. immunis*, which provided a detailed description of their genomic characteristics as well as more accurate and extensive information for further studying the gene arrangement and the evolutionary history of Microgastrinae.

## 2. Materials and Methods

### 2.1. Sample Identification and DNA Extraction

Before DNA extraction, the obtained specimen was preserved in 100% ethanol at −80 °C. The specimens of two parasitoids were obtained through rearing in the laboratory from the Tea Research Institute, Chinese Academy of Agricultural Sciences. Following the manufacturer’s instructions, genomic DNA was extracted using FastPure Cell/Tissue DNA Isolation Mini Kit (Vazyme Biotech Co., Ltd., Nanjing, China). The voucher specimen was preserved in the Parasitic Hymenoptera Collection (Institute of Insect Sciences, Zhejiang University).

### 2.2. Next-Generation Sequencing and Assembly

The library was created using the VAHTS™ Universal DNA Library Prep Kit for Illumina^®^ v9.1, and sequenced on an Illumina HiSeq platform with 150 bp pared-end read length by Novogene. FastQC was utilized to examine the raw data, and Trimmomatic was utilized with default settings to trim adaptors and indexes [18,19]. The FastqExtract script was used to filter out the target mitochondrial reads by running BLASTn (E value: 1 × 10^−5^) against a reference data set of Braconidae mitochondrial genomes [20]. The mitochondrial reads were assembled by SPAdes v3.0 [21] and IDBA v1.1.3 with default values, respectively [22]. Following that, GENEIOUS Prime v2020.0.5 (Biomatters Ltd., San Diego, CA, USA) was used to integrate two assemblies.

### 2.3. Mitochondrial Genome Annotation and Analysis

The MITOS Web Serve was used to annotate the assembled genomes [23]. Against the reported mitogenomes of Braconidae, the start and stop positions of protein-coding genes were manually modified in Geneious Prime v2020.0.5. The predicted tRNA genes were verified by the tRNAscan-SE search site with their homologs from related species [24]. Two maps of mitochondrial genomes were created by CGView server online V 1.0 (http://cgview.ca/, accessed on 25 February 2022) [25]. The obtained mitogenomes were uploaded to GenBank (OP741148 and OP741149).

Gene rearrangements were investigated by contrasting them with the putative ancestral mitogenome of *Drosophila melanogaster*. The base composition of all components was estimated using MEGA 11.0 [26]. The following formulas were used to calculate AT-skew and GC-skew: AT-skew = (A% − T%)/(A% + T%) and GC-skew = (G% − C%)/(G% + C%) [27]. Geneious Prime v2020.0.5 was used to calculate the relative synonymous codon usage (RSCU) of all PCGs. DnaSP v6.12.03 was used to compute the Synonymous (Ks) and non-synonymous (Ka) substitution rates of PCGs [28].

### 2.4. Phylogenetic Analysis

G-INS-i methods implemented in MAFFT v7.464 were used to align the PCGs [29]. PartitionFinder v2 was used to find the optimal partition schemes of substitution models for the matrix, with model selection = BIC and Branch lengths = unlinked across different subsets [30]. Based on nucleotide sequences of all PCGs, Mrbayes v3.2.7a [31] and RAxML-HPC2 v8.2.12 [32] were used to construct the phylogenetic tree, respectively. Four Markov chains were run simultaneously for 10 million generations for Bayesian inference analysis (BI), with tree sampling occurring every 1000 generations and a burn-in of 25% of the trees for phylogenetic analysis. In maximum likelihood (ML) analysis, ML trees with 1000 bootstrap replications are built using the GTRGAMMA model and 200 runs for various individual partitions.

## 3. Results

### 3.1. General Features of Mitochondrial Genomes

The mitochondrial genomes of *Pa. hyposidrae* and *Pr. immunis* are 17,063 bp and 16,397 bp in length, respectively (Table 1, Figure 1). 37 mitochondrial genes were identified in *Pa. hyposidrae* and *Pr. immunis*, containing 13 PCGs, 22 tRNA genes, and 2 ribosomal RNA (rRNA) genes. The complete control region of the two species was unable to be assembled, probably owing to poor similarity among reference sequences and high duplications of A and T, which are typical in hymenopteran mitogenomes [33].

There are six overlapping regions in the mitochondrial genome of *Pa. hyposidrae*, with a total of 23 bp, as the largest overlap was 7 bp, located at two junctions (*trnW-trnC, trnS2-nad1*) and the smallest overlap was 1 bp, located at one junction (*nad2-trnW*). There are 21 intergenic regions with a total length of 2271 bp, as the largest length of the gene spacer was 1129 bp, located at one junction (*trnP-trnT*) and the smallest length of the gene spacer was 1 bp, located at two junctions (*trnQ-nad2, cox2-trnH*). The length of *rrnL* and *rrnS* is 1127 bp and 751 bp, respectively, and the length of tRNA is 54–74 bp.

There are three overlapping regions in the mitochondrial genome of *Pr. immunis* with a total of 18 bp, as the largest overlap was 10 bp, located at one junction (*atp8-atp6*) and the smallest overlap was 1 bp in length, located at one junction (*trnS1-trnA*). There are 29 intergenic regions with a total length of 1849 bp, as the largest length of the gene spacer was 579 bp, located at one junction (*nad5-trnS1*) and the smallest length of the gene spacer was 1 bp, located at three junctions (*trnQ-nad2, nad4L-trnT, trnP-ad6*). The length of *rrnL* and *rrnS* is 1137 bp and 661 bp, respectively, and the length of tRNA is 60–73 bp.

All PCGs in *Pa. hyposidrae* and *Pr. immunis* mitochondrial genomes started with a typical ATN codon, as in other Hymenoptera insects [34,35,36]. Among the starting codons used by the mitochondrial genome of *Pa. hyposidrae*, ATT was used the most, seven times, followed by ATG four times, and ATA only twice. All protein coding genes except *cox3* use T as a termination codon, and the other twelve protein coding genes use TAA. Incomplete termination codons often exist in the protein coding genes of the mitochondrial genome of arthropods. After being transcribed into mRNA, they are supplemented by 3 ‘end polyadenylation [37,38]. Among the start codons used by the mitochondrial genome of *Pr. immunis*, ATG was used the most, six times, followed by ATT five times, and ATA only twice. All protein coding genes use TAA as the termination codon.

### 3.2. Base Composition, and Codon Usage

The sequenced region of the mitogenomes of *Pa. hyposidrae* and *Pr. immunis* have 86.15% and 86.56% A + T content, respectively (Table 1). Compared with other orders, the relatively high A + T content in the hymenopteran mitogenomes is not exceptional [39]. The total A + T content of all PCGs of *Pa. hyposidrae* and *Pr. immunis* is 85.12% and 85.52%, respectively. Typically, *nad6* or *atp8* has the largest proportion of A + T content in the hymenopteran mitochondrial genome [40]. In the mitochondrial genomes of *Pa. hyposidrae* and *Pr. immunis*, the highest A + T content is *atp8* (*Pa. hyposidrae*: 94.55%, *Pr. immunis*: 93.10%), and the lowest A + T content is *cox1* (*Pa. hyposidrae*: 77.30%, *Pr. immunis*: 77.71%) (Figure 2). In *Pa. hyposidrae*, the A + T content of *nad6* is 92.99% while the A + T content of *nad6* is 91.67% in *Pr. immunis*. Most of the AT skew is negative in both *Pa. hyposidrae* and *Pr. immunis*. In *Pa. hyposidrae*, the largest AT skew is *nad4* (0.1202), and the lowest is *atp6* (−0.2509). In *Pr. immunis*, the largest AT skew is *nad3* (0.1485), and the lowest is *atp6* (−0.2076). Similarly, the GC skew also has positive and negative values. Most of the GC skew is positive in both *Pa. hyposidrae* and *Pr. immunis*. In *Pa. hyposidrae*, the largest GC skew is *cox2* (0.2137), and the smallest is *atp8* (−0.3333). In *Pr. immunis*, the largest GC skew is *nad2* (0.1959), and the smallest is *nad3* (−0.2223).

The relative synonymous codon usage (RSCU) in the mitochondrial genomes of *Pa. hyposidrae* and *Pr. immunis* displayed a strong preference toward A and T, particularly at the third position. The total number of codons in the *Pa. hyposidrae* and *Pr. immunis* mitochondrial genomes is 3751 and 3678, respectively. The four most frequent used amino acids in both *Pa. hyposidrae* and *Pr. immunis*, correspond to the following codons: UUA (Leu), UUU (Phe), AUU (Ile), and AUA (Met) (Appendix A). These findings are similar with reported mitogenomes of other hymenopterans [41,42].

### 3.3. Gene Rearrangements

Gene rearrangement events are typically divided into four types: translocation, gene shuffling (local translocation), local inversion (inverted in the local position), and remote inversion (translocated and inverted) [11]. Compared with the putative ancestral mitogenome of *D. melanogaster*, the PCGs and tRNAs have various degrees of rearrangement in the mitogenomes of the two parasitoid wasps (Figure 3).

Frequent gene rearrangements have been reported in Braconidae in previous research. The majority of genes rearranged were tRNA genes, with three hotspots for tRNA gene rearrangement proposed: *trnA-trnR-trnN-trnS1-trnE-trnF*, *trnK-trnD* and *trnT-trnP* [15]. In the two species, *Pa. hyposidrae* and *Pr. Immunis*, the three hot spots for tRNA gene rearrangement are also discovered. *trnH* is remotely inverted to the junction of *cox2* and *atp8*, forming the organization pattern *trnH*-*trnD*-*trnK* in *Pa. hyposidrae*, and *trnH*-*trnK*-*trnD* in *Pr. Immunis*. *trnA-trnR-trnN-trnS1-trnE-trnF* is rearranged into *trnN-trnE-trnA-trnS1* and *trnS1-trnA-trnE-trnN* in *Pa. hyposidrae* and *Pr. Immunis*, respectively. *trnT*-*trnP* is remotely inverted from downstream to upstream of *nad4L* in *Pa. hyposidrae*, forming the arrangement pattern *trnP*-*trnT*. Additionally, the clusters *trnW-trnC-trnY* and *trnI-trnQ-trnM* are also rearranged. Both species have locally inverted *trnY*. *trnI* and *trnM* were remotely inverted, resulting in the arrangement pattern *trnM*-*trnI*-*trnQ* in *Pa. hyposidrae*, and *trnI*-*trnM*-*trnQ* in *Pr. Immunis*.

With the exception of *Chelonus* sp., *Cotesia vestalis*, and *Stenocorse bruchivorus*, all protein coding genes in all Braconidae species were conserved as the putative ancestral arrangement of insects [16]. The rearrangement of protein coding genes was identified in *Pa. hyposidrae* and *Pr. Immunis*. *Nad4L* is locally inverted in *Pa. hyposidrae* and a large block (*cox3-nad3-nad5-nad4*) is remotely inverted to *nad4-nad5-nad3-cox3* in *Pr. Immunis*, both of which are unique gene rearrangement patterns in the Braconidae.

### 3.4. Phylogenetic Analyses

To validate the phylogenetic position of *Pa. hyposidrae* and *Pr. Immunis* within Microgastrinae, a phylogenetic analysis based on all PCGs was constructed in this study using Bayesian and maximum likelihood inferences with other 14 organisms, including 12 species from the subfamily Microgastrinae (the tribe concepts follows a classification based on Manson [43], which is widely used) and two from the helconoid subfamily complexes (Macrocentrinae and Agathidinae) as outgroup (Figure 4, Appendix A). The topology of BI tree is the same as that of ML tree, though some clades with low support values, which supports that the subfamily Microgastrinae is a monophyletic group, and is a sister group to the helconoid subfamily complexes [15,44]. Apantelini and Cotesiini split into two and three clusters, separately, indicating they are paraphyletic, which is supported by some previous studies [2,4]. The results of this study show that *Pa. hyposidrae* and *Pr. Immunis* belong to Apantelini and Cotesiini, respectively. Several genera with unclear boundaries, as Fernandez-Triana et al. pointed out, including *Protapanteles* and *Diolcogaster*, are most likely polyphyletic and will need to be split into several genera [3]. *Pr. Immunis* is sister to *C. vestalis* and *Cotesia flaps* rather than *Protapanteles* sp., indicating that the genera *Protapanteles* is paraphyletic.

## 4. Conclusions

In this study, we successfully obtained the two mitogenomes of *Pa. hypsidrae* and *Pr. Immunis*, which are the most important parasitic wasps of *E. grisescens* and *E. obliqua*. Combined with reported Braconidae mitogenomes, we conducted analyses of base composition, codon usage, gene rearrangement, and phylogeny within Braconidae. Two novel gene rearrangement types were discovered in the two newly acquired mitogenomes, the local inversion of *nad4L* in *Pa. hyposidrae* and the remote inversion of the block *nad4-nad5-nad3-cox3* in *Pr. Immunis*. The phylogenetic analysis supported the subfamily Microgastrinae is a monophyletic group, but the tribes Apantelini and Cotesiini within this subfamily are paraphyletic groups. Nevertheless, it is crucial to grasp these results in order to comprehend the evolution of Microgastrinae. For further research into gene organization and the evolutionary history of Microgastrinae, denser taxon sampling yields more accurate and thorough data.

## Figures and Tables

**Figure 1 genes-14-00230-f001:**
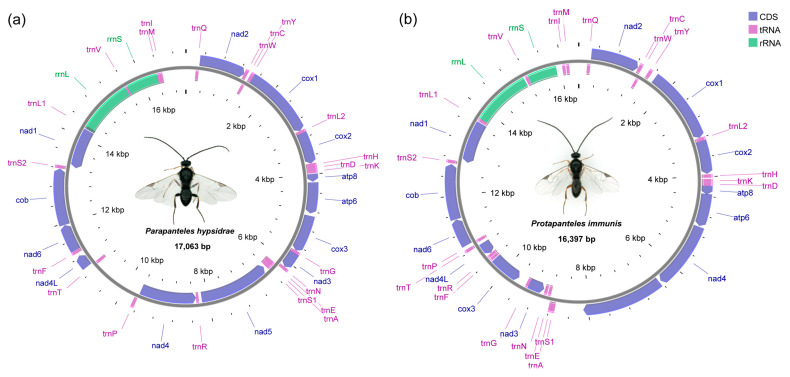
Maps of the mitochondrial genomes of *Pa. hypsidrae* (**a**), and *Pr. immunis* (**b**). The circle shows the gene map (PCGs, rRNAs, and tRNAs). The genes shown outside the map are encoded on the majority strand (J strand), while the genes represented within the map are encoded on the minority strand (N strand).

**Figure 2 genes-14-00230-f002:**
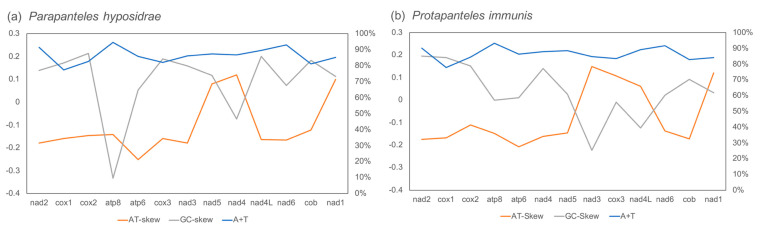
Base composition of protein-coding genes. (**a**) *Pa. hypsidrae*; (**b**) *Pr. immunis*.

**Figure 3 genes-14-00230-f003:**
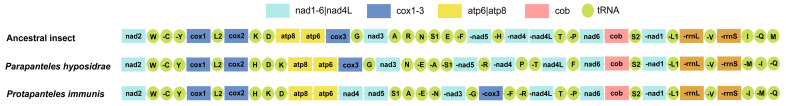
Gene rearrangement in the mitochondrial genome of *Pa. hypsidrae* and *Pr. immunis*.

**Figure 4 genes-14-00230-f004:**
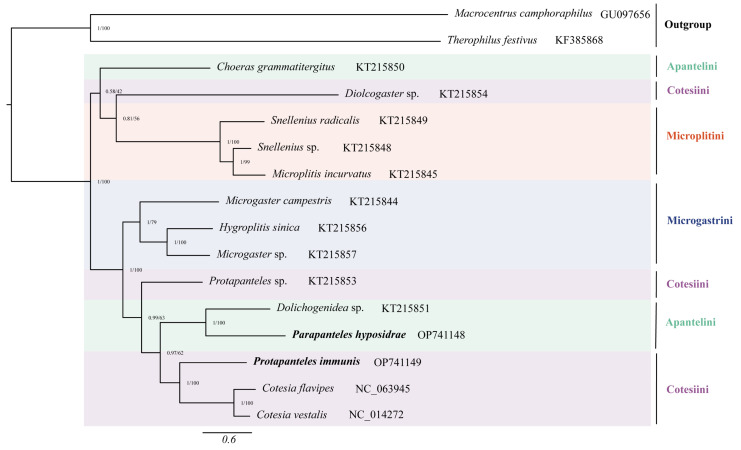
Phylogenetic analyses of Microgastrinae and two outgroups based on nucleotide datasets of 13 PCGs. Numbers separated by a slash on the node represent posterior probability (PP) (left) and bootstrap value (BV) (right). The graphic depicts the GenBank accession numbers for all species.

**Table 1 genes-14-00230-t001:** Base composition of *Pa. hypsidrae* and *Pr. immunis.*

Species	Whole Genome		Protein-Coding Genes			
	Length (bp)	A + T (%)	Length (bp)	A + T (%)	AT-Skew	GC-Skew
*Pa. hypsidrae*	17,063	86.15	11,311	85.12	−0.0696	0.1266
*Pr. immunis*	16,397	86.56	11,088	85.52	−0.1040	0.0847

## Data Availability

The data supporting the findings of this study are openly available in the National Center for Biotechnology Information (https://www.ncbi.nlm.nih.gov), accession numbers: OP741148 and OP741149. Raw sequence reads for each specimen-specific library were deposited in the BioProject PRJNA896708. The SRR codes for the two samples are SRR22142645 and SRR22142646.

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
