# Peer review of "The Mitochondrial Genomes of Two Parasitoid Wasps Protapanteles immunis and Parapanteles hyposidrae (Hymenoptera: Braconidae) with Phylogenetic Implications and Novel Gene Rearrangements"

_genes, 2023, doi:10.3390/genes14010230_

Round 1
Reviewer 1 Report
Dear authors
Congratulations! You have done a great work on the mitochondrial genomes of two parasitoid wasps Protapanteles immunis and Parapanteles hyposidrae. However, I am not agree with the statement on the phylogenetic status of Microgasrinae because only several species (representative from several tribe) included for the analysis. Therefore, statement related to that need to be deleted.
The other thing is about the discussion on both species. Please highlight on the novel gene on both. I think that information inadequate.
1. Abstract- The phylogenetic analysis supported the subfamily Microgastrinae is a monophyletic group, but the tribes Apantelini and Cotesiini within this subfamily are paraphyletic groups.- Try to avoid this statement. This only constructed based on several species, then not strong to say the mono or poly status of the Microgastrinae.
2. Results. Point 3.4 Delete
3. Fig. 4 delete
4. Line 239-242 delete
5. Line 360, Apanteles italic
6. Line 229-280 f Ectropis 279 obliqua and Ectropis grisescens - italic
7. Introduction- I think information on both species (biology, behavior, morphology need to brief explain under Introduction). The significant to conduct the gene arrangement on both species.
8. Line 342 Diadegma semiclausum - italic
9. Line 296 Hycleus genera - italic and genus?
10. Line 291 Heterodoxus macropus - italic
11. Line 285 Drosophila melanogaster - italic
12. Line 281 Bothriometopus italic
Thank you.
Author Response
Point 1: Abstract- The phylogenetic analysis supported the subfamily Microgastrinae is a monophyletic group, but the tribes Apantelini and Cotesiini within this subfamily are paraphyletic groups.- Try to avoid this statement. This only constructed based on several species, then not strong to say the mono or poly status of the Microgastrinae.
Response 1: Thank you very much for your suggestions. We acknowledge that there are some limitations in the sampling scope of this study, and increasing the sampling scope is one of the research directions in the future. The data we used for phylogenetic analysis have included all available data of mitochondrial genomes of the subfamily Microgastrinae, and have included four of the five widely accepted tribes in this subfamily. Our phylogenetic analysis explored the phylogenetic relationships within the subfamily Microgastrinae for the first time based on the mitochondrial genomes of Microgastrinae, including the two newly sequenced. Our conclusions revealed that these two newly sequenced species belong to their corresponding tribe as the morphological study supported on the one hand, and on the other hand, our results confirmed the results of phylogenetic analyses of this subfamily of previous researches. We stated all these in the original text of our paper. Therefore, we thought that this conclusion could be retained.
Please also see the two articles that we already cited in our original manuscript:
Banks, J.C.; Whitfield, J.B. Dissecting the ancient rapid radiation of microgastrine wasp genera using additional nuclear genes. Mol. Phylogenet. Evol. 2006, 41, 690-703.
Dowton, M.; Austin, A.D. Phylogenetic Relationships among the Microgastroid Wasps (Hymenoptera: Braconidae): Combined Analysis of 16S and 28S rDNA Genes and Morphological Data. Molecular Phylogenetics & Evolution 1998, 10, 354.
Point 2: Results. Point 3.4 Delete.
Response 2: As mentioned in Response 1, we thought that this part was worth being kept.
Point 3: Fig. 4 delete.
Response 3: As mentioned in Response 2, we thought the picture was also worth being kept.
Point 4: Line 239-242 delete.
Response 4: As for the answer to the first question, we thought this part could be kept.
Point 5: Line 360, Apanteles italic.
Response 5: We have revised this.
Point 6: Line 229-280 f Ectropis 279 obliqua and Ectropis grisescens – italic.
Response 6: We have revised this.
Point 7: Introduction- I think information on both species (biology, behavior, morphology need to brief explain under Introduction). The significant to conduct the gene arrangement on both species.
Response 7: We have added relevant descriptions of behavior and morphology in the second paragraph of the introduction. In addition, we have added the significance of conducting the gene arrangement on both species in the fourth paragraph.
Point 8: Line 342 Diadegma semiclausum - italic.
Response 8: We have revised this.
Point 9: Line 296 Hycleus genera - italic and genus?
Response 9: We have revised this.
Point 10: Line 291 Heterodoxus macropus – italic.
Response 10: We have revised this.
Point 11: Line 285 Drosophila melanogaster - italic.
Response 11: We have revised this.
Point 12: Line 281 Bothriometopus italic.
Response 12: We have revised this.
Reviewer 2 Report
Xiao et al., assembled the mitochondorial DNA of two parasitoid wasps Protapanteles immunis and Parapanteles hyposidrae by utilizing NGS, and performed phylogenetic analysis. The authors found some gene arrangements.
The results are clarely presented and the logic is mostly easy to be followed. I have a few comments for potential improvements.
L38 I don't understand "it is likley that ...". Can you rephrase the sentence?
L106 You used nucleotides for phylogenetic analysis. Can you also run the programs using translation because sometimes translation make the more robust phylogeny than nucleotides do.
Author Response
Please see our response attached.
